# OpenReview forum: "Steering Multimodal Large Language Models Decoding for Context-Aware Safety"
_ICLR.cc/2026/Conference — ICLR 2026 Conference Withdrawn Submission_

### Official Review · Reviewer_DXhW · 2025-10-22

**Soundness:** 2
**Presentation:** 3
**Contribution:** 3
**Rating:** 4
**Confidence:** 4

**Summary:**

The paper proposes SafeCoDe, a lightweight and model-agnostic decoding framework to improve context-aware safety in MLLMs.
The method addresses two common issues: undersensitivity to visually implied risks and oversensitivity causing unnecessary refusals.
SafeCoDe operates in two stages: (1) constrastive decoding, which highlights tokens sensitive to visual context by contrasting real and noised images, and (2) global-aware token modulation, which integrates a global safety verdict from GPT-4o to adapt refusal behaviors according to the scene.
Experiments on multimple MLLMs and safety benchmarks show that SafeCoDe consistently improves context-sensitive refusal accuracy while preserving general task utility.

**Strengths:**

- This paper is very well written, with intuitive figures that clearly illustrate the key ideas and visual examples that highlight context-aware safety scenarios. The methodology and experimental design are easy to follow.
- This paper introduces a novel inference-time decoding framework (SafeCoDe) for multimodal safety alignment, which requires no model retraining. SafeCoDe combines constrastive visual decoding and global-aware token modulation to jointly address oversensitivity, which provides a creative and practical perspective on safety steering through token-level modulation grounded in visual context.
- The evaluation setup is rigorous, and results convincingly show SafeCoDe's effectiveness and robustness.

**Weaknesses:**

1. SafeCoDe's effectiveness depends heavily on the external safety judge. Although the authors include some ablations in App. G and App. I, the results show noticeable variance across different judges. For example, performance on MOSSBench with Qwen even drops below the base model. This suggests that SafeCoDe's robustness and generality remain limited. Relying on GPT-4o as the primary judge also raises cost and scalability concerns, which may hinder real-world applicability. Exploring lighter or self-contained judging mechanisms could make the approach more practical and stable.
2. The interpretation of Tab. 1 seems overstated. The authors conclude that MLLMs ''anchor their refusals primarily on textual input while neglecting visual contexts,'' based on similar results between real and blank images. However, this experiment may not fully support that claim, since benchmarks like MSSBench and MOSSBench may already contain safety-related textual cues. Th results could therefore reflect dataset bias rather than genuine unimodal over-reliance. To better substantiate this point, the authors could include tasks where safety depends mainly on visual cues (e.g., MSTS[1]-style prompts such as ``Should I do this?''), where textual information alone is insufficient for correct judgment.
3. The construction of the refusal token space is rather heuristic and brittle. Since the tokenizer typically splits a phrase like ''I'm sorry, but...'' into multiple subword tokens, using a string-based list to identify ''refusal tokens'' may be unreliable and prone to false positives or partial matches. It is also unclear how multi-token partterns are consistently detected during decoding. Moreover, the handcrafted list seems incomplete. For example, common safety-prefixed tokens such as ''understand'' and ``note that'' are not covered, making the modulation potentially inconsistent across models. In addition, SafeCoDe applies the modulation only within the first 2-5 decoding steps, which likely enforces only shallow safety alignment[2]. While this avoids over-regularization, it limits the framework's ability to sustain safety intent throughout generation.

[1]. Röttger, Paul, et al. "MSTS: A Multimodal Safety Test Suite for Vision-Language Models." arXiv preprint arXiv:2501.10057 (2025).

[2]. Qi, Xiangyu, et al. "Safety alignment should be made more than just a few tokens deep." arXiv preprint arXiv:2406.05946 (2024).

**Questions:**

Q1: The claim that current MLLMs ''anchor their refusals primarily on textual input'' seems undersupported, since MSSBench and MOSSBench both include safety-related textual cues. Could you clarify whether the dataset bias may confound this conclusion?

Q2: How exactly is the refusal token space is implemented, given that tokenizers split phrases like ''I'm sorry'' into multiple sub-tokens? Have you evaluted the coverage or false-positive rate of this handcrafted list across different models?

Q3: It would be useful to understand whether shallow early-step intervention limits the persistence of safey alignment during generation.

Q4: Considering that GPT-4o is expensive to call repeatedly during inference, do you foresee a path to make SafeCoDe deployable in practice (e.g., via distilled or self-contained judges)? Any preliminary results in this direction would strengthen the paper’s applicability.

---

### Official Review · Reviewer_TtsJ · 2025-10-23

**Soundness:** 2
**Presentation:** 3
**Contribution:** 2
**Rating:** 2
**Confidence:** 4

**Summary:**

Multimodal LLMs often struggle with context-aware safety, tending toward oversensitivity (unwarranted refusals) or undersensitivity (missed visually grounded risks).  This paper introduces SafeCoDe, a lightweight, model-agnostic decoding framework that dynamically adjusts token generation based on multimodal context. SafeCoDe (1) contrasts real and Gaussian-noised images to surface visually sensitive tokens, and (2) integrates scene-level reasoning with token-level modulation to calibrate refusals according to a predicted safety verdict. Across diverse architectures and safety benchmarks, SafeCoDe consistently improves context-sensitive refusals while preserving helpfulness.

**Strengths:**

- The method is intuitive and directly targets the model’s limitations, with clear logic.
- The paper is well written and well structured. The problem, method, and experiments are clearly explained.
- The proposed method clearly outperforms all baselines in the experiments.

**Weaknesses:**

- Section 1.1 is titled “KEY OBSERVATIONS AND INSIGHTS,” but it mostly repeats known points from prior work: Over-reliance on Textual Modality and Lack of Global Information. The authors do not provide new insights. Also, the evaluations in Section 1.1 are all done on open-source models with fewer than 10B parameters, which makes me unsure how reliable the conclusions are.
- The proposed method relies on a powerful model (e.g., GPT-4o, mentioned in Section 2.2) to obtain a global safety verdict. Since the later experiments are all on open-source models under 10B parameters (much weaker than GPT-4o), directly comparing the proposed method with the baselines seems unfair.
- The baselines are too weak and all prompt-based. Stronger baselines should be added, e.g., [1], [2], [3]. Like this paper’s method, these baselines also do not require training the original model.

[1] Wang H, Wang G, Zhang H. Steering away from harm: An adaptive approach to defending vision language model against jailbreaks[C]//Proceedings of the Computer Vision and Pattern Recognition Conference. 2025: 29947-29957.

[2] Ghosal S S, Chakraborty S, Singh V, et al. Immune: Improving safety against jailbreaks in multi-modal llms via inference-time alignment[C]//Proceedings of the Computer Vision and Pattern Recognition Conference. 2025: 25038-25049.

[3] Yilei Jiang, Xinyan Gao, Tianshuo Peng, Yingshui Tan, Xiaoyong Zhu, Bo Zheng, and Xiangyu Yue. 2025. HiddenDetect: Detecting Jailbreak Attacks against Multimodal Large Language Models via Monitoring Hidden States. In *Proceedings of the 63rd Annual Meeting of the Association for Computational Linguistics (Volume 1: Long Papers)*, pages 14880–14893, Vienna, Austria. Association for Computational Linguistics.

**Questions:**

1. Could you explain how you chose the baselines and why this choice is reasonable?
2. Why were all the evaluation experiments run only on open-source models with fewer than 10B parameters?
3. How do you view the fact that the method proposed in this paper depends on a powerful model (GPT-4o)?

---

### Official Review · Reviewer_fzAm · 2025-10-28

**Soundness:** 3
**Presentation:** 3
**Contribution:** 2
**Rating:** 2
**Confidence:** 4

**Summary:**

The paper studies context‐aware safety for MLLMs, aiming to reduce both undersensitivity and oversensitivity issue for safety-related queries. To this end, the authors propose SafeCoDe, a decoding approach with two stages: (i) contrastive decoding, which surfaces tokens sensitive to visual context by contrasting logits from the real versus Gaussian-noised image; and (ii) global-aware token modulation, which uses an auxiliary MLLM judge to derive a scene-level safe/unsafe verdict and softly boost or suppress refusal-token logits during the first few decoding steps. Extensive experiments across four open MLLMs and multiple safety benchmarks show that SafeCoDe improves the balance between safety and helpfulness while preserving general utility, outperforming evaluated baselines. The authors acknowledge limitations including the need for access to model logits and the resulting issue of overly hard refusals.

**Strengths:**

- **Broad empirical coverage and strong results:** The paper evaluates four backbone MLLMs across diverse benchmarks, including contextual safety, general safety, and **utility** tasks, together with seven implemented baseline methods, demonstrating the comprehensiveness of the experimental study. The proposed method consistently outperforms the compared baselines by a clear margin.
- **Comprehensive ablations and thorough appendix:** The paper analyzes the contribution of each module in the proposed method through detailed ablation studies and further examines the impact of different model choices for the MLLM judge, forming a rigorous and systematic evaluation. The appendix provides concrete implementation details, extended results, and illustrative case studies, which enhance the transparency and credibility of the work.
- **Clear presentation:** The paper is well organized, clearly written, and easy to follow.

**Weaknesses:**

- **Method Design: dependency on an external MLLM judge.** SafeCoDe relies on a separate *MLLM judge,* set as **GPT-4o,** to provide the global safety verdict, essentially outsourcing the "global information" collection to another model. From a conceptual standpoint, this somewhat undermines the objective of improving MLLMs' safety context awareness (if access to GPT-4o is assumed, why are we still evaluating these weaker models?). As shown in Table 7, replacing GPT-4o with a smaller Qwen2.5-3B judge notably weakens performance, especially on oversensitivity cases. Moreover, using an external proprietary model may lead to an unfair comparison with baselines; in fact, another insightful baseline here could be the GPT-4o itself.
- **Method Efficiency: unverified "lightweight" claim.** Although the authors describe SafeCoDe as *lightweight*, no empirical evidence (e.g., latency, FLOPs, or cost) supports this claim. The method involves contrastive decoding (requiring two forward passes) and an additional GPT-4o query, both of which can substantially increase inference overhead. An efficiency or runtime comparison with baseline methods would help substantiate the claim and clarify real-world practicality.
- **Evaluation: reliance on hyperparameters and configurations.** SafeCoDe depends on several tunable components, such as the five hyperparameters listed in Table 6, decoding temperature (unspecified), and refusal-string lists (Table 5), that may affect performance. This raises concerns about robustness and reproducibility across models and datasets, as well as the cost of hyperparameter tuning. Additional experiments on parameter sensitivity or a clearer description of the selection heuristic would strengthen the evaluation.

**Questions:**

1. How were the hyperparameters in Table 6 determined?
2. Why is AdaShield-adaptive not included for the AdaShield baseline implementation?
3. Given the use of an external GPT-4o judge, have the authors tried a simpler method that directly appends the GPT-4o verdict to the user query without modifying token logits (as in global-aware token modulation stage)?

---

### Official Review · Reviewer_sRdd · 2025-10-30

**Soundness:** 3
**Presentation:** 3
**Contribution:** 3
**Rating:** 4
**Confidence:** 3

**Summary:**

This paper addresses the critical challenge of context-aware safety in Multimodal Large Language Models (MLLMs). The authors identify two primary failure modes: undersensitivity, where models fail to refuse harmful requests in unsafe visual contexts, and oversensitivity, where they unnecessarily reject benign queries. To tackle this issue, the paper introduces SafeCoDe (Safety-aware Contrastive Decoding), a lightweight, model-agnostic, inference-time decoding framework. Through extensive experiments on four different MLLMs and a wide range of safety and utility benchmarks, the authors demonstrate that SafeCoDe consistently improves the balance between undersensitivity and oversensitivity, outperforming several baseline methods while largely preserving the model's general-purpose capabilities.

**Strengths:**

- The paper clearly articulates the critical yet often overlooked duality of undersensitivity and oversensitivity in MLLM safety. This framing is insightful and provides a more complete picture of the challenges in multimodal alignment. The examples provided are highly effective at motivating the work.

- SafeCoDe is an elegant and well-designed framework. The two-stage approach is logically sound: the contrastive decoding stage grounds the model in visual specifics, while the global modulation stage enforces a high-level safety decision. This design directly addresses the identified weaknesses of prior methods (unimodal bias and lack of global context).

- The experimental evaluation is a major strength of this work. The authors are thorough in their choice of base models, relevant baselines, and a wide array of benchmarks. Testing across contextual safety, general safety/jailbreaking, and model utility provides a convincing and holistic validation of their claims.

- The proposed method is an inference-time technique that does not require model retraining, making it broadly applicable to many existing MLLMs. This is a significant practical advantage.

**Weaknesses:**

- The primary weakness is the framework's dependency on an external, powerful, and proprietary model (GPT-4o) to act as the safety judge. This has several negative implications: (1) Calling a large model like GPT-4o for every inference request introduces significant latency and computational/monetary cost, which contradicts the paper's claim of being a "lightweight decoding framework." The decoding part might be lightweight, but the entire pipeline is not. (2) Reliance on a black-box API makes the results difficult to reproduce perfectly and limits the method's accessibility to researchers without API access. (3) The comparison with baselines that are self-contained seems unfair. The impressive performance of SafeCoDe may be disproportionately due to the superior reasoning ability of GPT-4o. The paper would be much stronger if results using an open-source judge were featured in the main tables to provide a more grounded and fair comparison.

- As the authors acknowledge in their failure analysis, SafeCoDe often produces blunt, uninformative refusals (e.g., "I'm unable to help you... since it is unsafe"). While this prevents harm, it misses an opportunity to be truly helpful by explaining why the requested action is unsafe (e.g., "Placing a credit card in a microwave is dangerous as it can create sparks, damage the microwave, and release toxic fumes."). This is a limitation in achieving a truly aligned and helpful model.

**Questions:**

- Could you please elaborate on the decision to rely on GPT-4o for the main results? How do you justify the fairness of comparing your system, which includes this powerful external component, against self-contained baselines? To provide a fairer picture, would you be willing to add the results using the open-source Qwen judge to the main experimental tables?

- Could you quantify the additional latency and potential cost introduced by the MLLM judge per inference? How does this impact the claim that SafeCoDe is a lightweight framework, especially in real-world deployment scenarios?

- Your failure analysis insightfully points out that refusals can be overly blunt. Have you considered extending the framework to generate more informative or softer refusals? For example, could the global safety verdict be a more structured output (e.g., identifying the specific risk) that could then guide the generation of a helpful explanation alongside the refusal?

---

### Note · Authors · 2025-12-01

**Comment:**

We appreciate all of the reviewers’ comments and recommendations, which we found very useful and constructive. However, due to time constraints, we are unable to complete all of the additional experiments within the rebuttal period. We have taken all feedback seriously, and we agree that further analysis and additional experiments are necessary. We will incorporate all suggestions to further refine the paper, and we sincerely thank all reviewers and the area chairs for their efforts and insights.

**Withdrawal Confirmation:**

I have read and agree with the venue's withdrawal policy on behalf of myself and my co-authors.